# Exchangeability in Neural Networks and its Application to Dynamic Pruning

## Abstract

Modern neural networks (NN) contain an ever-growing number of parameters, substantially increasing the memory and computational cost of inference. Researchers have explored various ways to reduce the inference cost of NNs by reducing the model size before deployment and dynamically pruning the inference computation at runtime. In this work, we present ExPRUNE, a general, dynamic pruning optimization that enables multi-granularity partial computation on a per-input basis. ExPRUNE requires no change to the model architecture or the training algorithm. ExPRUNE is based on our theoretical results that the relationship between certain model parameters and intermediate values can be described by a statistical property called *exchangeability*. By identifying exchangeable parameters and values in the model, we are able to first partially evaluate the network, analyze the statistics of the partial results, and make pruning decisions on the fly. Because ExPRUNE is theory grounded, it generalizes across model architectures in different problem domains. We evaluate ExPRUNE on one computer vision models, one graph model and one language model. ExPRUNE provides 10.98–17.33% reduction in FLOPs with negligible accuracy drop and 21.61–27.16% reduction in FLOPs with at most 1% accuracy drop. We also demonstrate that ExPRUNE composes with static magnitude pruning. On models that have been aggressively statically pruned, ExPRUNE still provides additional 10.24–11.11% reduction in FLOPs with negligible accuracy drop and 13.91–14.39% reduction in FLOPs with at most 1% accuracy drop.

## 1 Introduction

Modern neural networks (NN) contain an ever-growing number of parameters, substantially increasing the memory and computational cost of inference (Han et al., 2022). To tame resource usage, researchers have developed a number of NN optimizations that statically reduce model size before deployment, including static pruning (Cheng et al., 2024), quantization (Saha et al., 2024), knowledge distillation (Gou et al., 2021), and neural architecture search (Elsken et al., 2019). To a lesser extent, researchers have also developed optimizations that dynamically prune the inference computation (Teerapittayanon et al., 2016; Elkerdawy et al., 2022). Dynamic pruning is a runtime optimization that allows parts of the inference computation to be skipped on a per-input basis, enabling partial computation and improving performance. Prior dynamic pruning methods are usually specialized to specific model architectures and network granularities, and often require changes of the model architectures and the training algorithms. These limitations make them harder to apply to a broad class of models and training strategies.

In this work, we present ExPRUNE, a general, dynamic pruning optimization that enables multi-granularity (e.g., neuron/kernel/layer) partial computation on a per-input basis. ExPRUNE requires no changes to the model architecture or the training algorithm, as it exploits structures already present in the model. Specifically, ExPRUNE capitalizes on the presence of *exchangeable* model parameters and intermediate values. Exchangeability is a statistical property that implies identical distribution and symmetric interdependence among random variables (Dean & Verducci, 1990). By formalizing model training as drawing a random model from a distribution of models with respect to random initialization, we prove that certain model parameters have exchangeable marginal distributions, and so do the intermediate values computed with them. The property of exchangeability enables us to first partially evaluate the network, analyze the statistics of the partial results, and prune some computation on the fly. Because the ExPRUNE is grounded in these theoretical results, it can generalize across model architectures. We identify exchangeable parameter/value patterns in a range of modern NNs, including Convolution Neural Networks (CNNs), Graph Neural Networks (GNNs) and transformer-based language models (LMs).

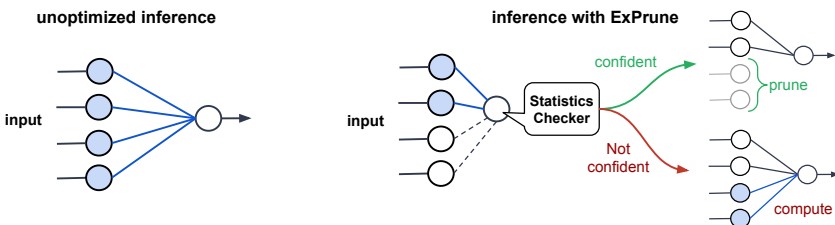

Figure 1: Dynamic pruning with EXPRUNE algorithm, grounded by our theory of exchangeability.

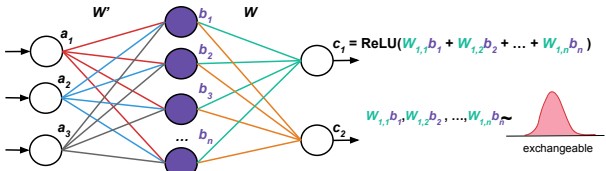
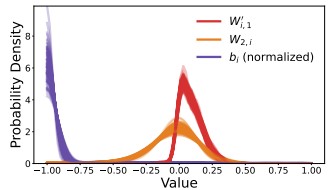

Figure 2: MLP without bias. $a,b,c$ are neuron activations. $W'$ and $W$ are weight matrices. Parameters and values with the same color are exchangeable, thus identically distributed.

Figure 3: Distributions of weights $W'_{i,1}$'s, $W_{2,i}$'s, and values $b_i$'s on one input over 500 trained models.

To demonstrate the efficacy of this method, we instantiate the EXPRUNE algorithm to prune two common structures present in NNs. We evaluate EXPRUNE on CNNs, GNNs, and LMs. We find that EXPRUNE can reduce the inference cost of these NNs by 10.98–17.33% with negligible drop on model accuracy. EXPRUNE is more general and outperforms the most similar prior work while requiring much less branching operations. We also empirically validate that EXPRUNE can compose with static pruning, providing additional efficiency improvements.

***Roadmap and Contributions:*** Section 2 presents a simple running example, providing intuitions of our theory and algorithm. Section 3 and Section 4 present the formalism and details. Section 5 presents our evaluation. We present the following contributions:

- *Analysis of statistical exchangeability in NNs* (Section 3): To our knowledge, this work is both the first to model the relationship among NN parameters with exchangeability and to perform dynamic pruning exploiting this property. The formalism provides insights into symmetry-induced redundancy identified by theoreticians (Lim et al., 2024).
- *Dynamic pruning algorithm* (Section 4): We present EXPRUNE, a general dynamic pruning algorithm based on our theoretical formulation. EXPRUNE can be applied to various NN architectures at various network granularities, and can compose with existing approaches for efficient inference. We also present two instantiations of EXPRUNE that prunes two common structures in NNs.
- *Evaluation* (Section 5): We evaluate EXPRUNE on CNNs, GNNs, and LMs. We demonstrate that EXPRUNE provides 10.98–17.33% reduction in FLOPs with negligible accuracy drop and 21.61–27.16% reduction in FLOPs with <1% accuracy drop. We also demonstrate that EXPRUNE composes with static magnitude pruning. On models that have been aggressively statically pruned, EXPRUNE provides additional 10.24–11.11% reduction in FLOPs with negligible accuracy drop and 13.91–14.39% reduction in FLOPs with at most 1% accuracy drop.

## 2 RUNNING EXAMPLE: DYNAMIC PRUNING OF RELU-ACTIVATED MLP

In this section, we show how we use EXPRUNE to dynamically prune (Figure 1) a simple ReLU-activated multi-layer-perceptron (MLP) without bias (Figure 2), and the theoretical groundings behind it. In this example, we focus on two hidden layers (colored) with weights $W'$ and $W$. Denote the neuron activations of sequential layers shown as $a, b, c$. We have $b = \text{ReLU}(W'a), c = \text{ReLU}(Wb) = \text{ReLU}(W\,\text{ReLU}(W'a))$.

***Dynamic pruning.*** Figure 1 shows an illustration of our EXPRUNE dynamic pruning algorithm. It prunes the computation at runtime on a per-input basis. Assume that we are currently computing $c_1 = \text{ReLU}(\sum_{i=0}^{n} W_{1,i} b_i)$ at inference. Instead of computing all $n$ terms $W_{1,i} b_i$ for $i \le n$, EXPRUNE first computes $k$ terms, and decides whether computation of other terms can be skipped based on these $k$ terms. In this example, if $\sum_{i=0}^{k} W_{1,i} b_i \ll 0$, we can predict that the sum of all $N$ terms is likely to

be negative, so $c_1$ likely equals 0, and we can prune the computation of the other terms. We can predict the negativity of the final sum by simply comparing the partial sum with a threshold, or using more sophisticated methods like conducting a statistical test on the computed terms. This dynamic pruning techniques saves multiplication operations, translating to reduced compute and data movement.

In the above optimization, we use a partial sum to estimate the result of the full sum. We therefore implicitly assume that *the terms $W_{1,i}b_i$'s can be regarded as samples drawn from the same distribution*. Only if this assumption holds can we infer information about the other terms from the computed $k$ terms, enabling us to approximate the full result with partial computation. We develop a theory, which proves that these terms have *exchangeable* distributions, a stronger condition than the above assumption, because it in addition entails that their statistical dependence among each other is symmetric.

***Training as drawing a random model.*** Our theory is grounded in a statistical view of training as a random process, where the statistical properties of parameters are derived over the distribution of trained models. At the beginning of training, parameters are randomly initialized – this serves as the source of randomness in the training algorithm. Each random instantiation of the initialized model yields a different trained model. By analyzing all possibilities, the trained model can be viewed as a sample from a distribution of models, with respect to the random initializations. The distribution of the trained models, and the marginal distribution of any parameter in the trained models, are highly complex and intractable to derive precisely. Nonetheless, we can derive that certain parameters have exchangeable marginal distributions, and so do values computed from them.

***Exchangeable parameters.*** We develop a theory that analyzes the structure of the model and find that certain weights have *exchangeable* marginal distributions. For the MLP example, we analytically derive that for a fixed $j$ and different $i$'s, $W'_{i,j}$'s have exchangeable distributions, and so do $W_{j,i}$'s (as shown in Figure 2). To illustrate, we empirically study the identical distribution property, a weaker property entailed by exchangeability in Figure 3. We train 500 NNs (2-layer MLPs with 128 hidden neurons that classify MNIST) from 500 random initializations, and plot over the 500 trained models, the distribution of each $W'_{i,1}$, and the distribution of each $W_{2,i}$, for 128 different $i$'s in Figure 3. We can clearly see that $W'_{i,1}$'s are identically distributed, and $W_{2,i}$'s also follow an identical distribution, which is different from the distribution of $W'_{i,1}$'s.

***Exchangeable values.*** If we know certain parameters have exchangeable distributions, the values computed from these parameters will also have exchangeable distributions, given any input. In the example (Figure 2), $b_i$'s have exchangeable distributions, and so do the terms $W_{1,i}b_i$'s. In Figure 3, we also plot the distribution of each $b_i$ normalized to $[-1,1]$ on a specific MNIST image over 500 trained models to show that $b_i$'s are identically distributed. A large density of the distribution is near the minimum because $b_i$'s are ReLU activated and often 0. Specifically, the theoretical result that different terms $W_{1,i}b_i$'s can be viewed as samples from the same distribution grounds the dynamic pruning algorithm EXPRUNE.

## 3    EXCHANGEABILITY IN NEURAL NETWORKS

We formally define and derive exchangeability properties in the trained model. Note that this section is a pure theoretical analysis and does not make any change to the training algorithm or NN architectures.

### 3.1    BACKGROUND: STATISTICAL EXCHANGEABILITY AND PARAMETER SPACE SYMMETRY

Let $P$ be a $n \times n$ permutation matrix for some positive integer $n$. The following definition of exchangeability is taken from Kuchibhotla (2020) (assuming the probability density function exists).

**Definition 1** (Exchangeability). *Suppose $\zeta = (\zeta_1,...,\zeta_n) \in \mathcal{X}^n$ is a vector of random variables, $\zeta_i$'s are exchangeable iff their joint probability density function $p(\zeta)$ is invariant to input permutations, i.e., $\forall$ permutation matrix $P$ and $\zeta_0 \in \mathcal{X}^n$, $p(P\zeta_0) = p(\zeta_0)$.*

Exchangeability is a stronger condition than identical distribution, as it in addition implies the symmetric interdependence among $\zeta_i$'s, but it is weaker than $iid$ (Chow & Teicher, 2003). The following theorem states a condition under which a transformation on exchangeable random variables preserves their exchangeability (Kuchibhotla, 2020; Dean & Verducci, 1990) (stronger condition).

**Theorem 1** (Exchangeability Preservation). *Let $\zeta = (\zeta_1,...,\zeta_n) \in \mathcal{X}^n$ be vector of exchangeable random variables. Fix a transformation $G : \mathcal{X}^n \to \mathcal{X}^n$. If $G$ is permutation equivariant, i.e., $\forall$ permutation matrix $P$ and $\zeta_0 \in \mathcal{X}^n$, $PG(\zeta_0) = G(P\zeta_0)$, then $G(\zeta)$ is also a vector of exchangeable random variable.*

Given a NN architecture with $N$ real-valued parameters, we denote the NN function parameterized by $\theta \in \mathcal{R}^N$ as $f_\theta : \mathcal{X} \to \mathcal{Y}$, where $\mathcal{X}, \mathcal{Y}$ are input and output spaces respectively. A parameter space symmetry of the NN is defined as follows (Lim et al., 2024).

**Definition 2** (Parameter Space Symmetry). *A function $\omega : \mathcal{R}^N \to \mathcal{R}^N$ is a parameter space symmetry if $f_{\omega(\theta)}(x) = f_\theta(x), \forall x \in \mathcal{X}, \theta \in \mathcal{R}^N$, i.e., $f_{\omega(\theta)}$ and $f_\theta$ are the same function for any parameters $\theta \in \mathcal{R}^N$.*

## 3.2 Exchangeable Parameters and Values in Neural Networks

In this section, we view the trained model as a random model with respect to random initializations, and thus each parameter is a random variable. We prove that certain groups of parameters $\zeta_i$'s are exchangeable, and so are values $\xi_i$'s computed with $\zeta_i$'s respectively. Below is the key insight of our proof.

***Key Insight.*** In the initial model, $\zeta_i$'s have exchangeable distributions, as most popular NN initialization schemes draw parameters from $iid$ distributions. Some pose additional constraints on orthogonality or unit variances (Saxe et al., 2013; Mishkin & Matas, 2015) that introduce dependence but do not break exchangeability. Therefore, if each training step, as a transformation on the random variables, preserves exchangeability of $\zeta_i$'s, $\zeta_i$'s have exchangeable distributions in the trained model.

Denote the parameters of interest as $\zeta = (\zeta_1, ..., \zeta_n)$, where $\forall i, \zeta_i \in \mathcal{R}^m, \zeta \in (\mathcal{R}^m)^n$. For simple notations, we use the same variable name to denote $mn$-long vector in $\mathcal{R}^{mn}$ or $n$-long vector of $m$-long vectors $(\mathcal{R}^m)^n$ with the same elements in the row-major order. Assume that $\theta = \theta' \oplus \zeta$, where $\theta'$ is other parameters. Define a function $\omega_P : \mathcal{R}^N \to \mathcal{R}^N$ as $\omega_P(\theta) = \theta' \oplus P\zeta$, where $P\zeta = \oplus_{i=1}^n (P\zeta)_i \in \mathcal{R}^{mn}$, for some permutation matrix $P$. In other words, $\omega_P$ permutes $\zeta_i$'s in $\theta$ with the permutation matrix $P$.

**Theorem 2** (Exchangeable Parameters). *Given an NN architecture with function $f_\theta$, assume that $\zeta_i$'s have exchangeable distributions in the initial model. If for any permutation matrix $P$, $\omega_P$ is a parameter space symmetry, then $\zeta_i$'s have exchangeable distributions in the trained model.*

*Proof.* Here we assume using gradient descent for supervised learning for simplicity, but the the proof easily generalizes to other optimization algorithms. As we are only interested in $\zeta_i$'s, we formalize a training step as a transformation on only the $\zeta_i$'s.[1] Denote the NN loss function as $L_\zeta : \mathcal{X} \times \mathcal{Y} \times \mathcal{R}^{N-mn} \to \mathcal{R}$, parameterized by $\zeta$. $L_\zeta(x, y, \theta') = \psi(f_\theta(x), y)$ for some metric function $\psi$ such as cross entropy. A training step $G_S : (\mathcal{R}^m)^n \to (\mathcal{R}^m)^n$ takes a training batch of $B$ samples $S \in (\mathcal{X} \times \mathcal{Y})^B$, and does a gradient descent step $G_S(\zeta) = \zeta - \gamma \sum_{(x,y) \in S} \nabla_\zeta L_\zeta(x, y, \theta')$, where $\gamma$ is the learning rate, and $\nabla_\zeta L_\zeta(x, y, \theta')$ is the gradient of $L_\zeta$ with respective to $\zeta$, evaluated at $(x, y, \theta')$.

According to Theorem 1, we only need to prove that each training step $G_S$ is equivariant with respect to any permutation $P$ of $\zeta_i$'s, i.e., $PG_S(\zeta) = G_S(P\zeta)$, for all $\zeta, \theta', S$. We have

$$PG_S(\zeta) = P(\zeta - \gamma \sum_{(x,y) \in S} \nabla_\zeta L_\zeta(x, y, \theta')) = P\zeta - \gamma \sum_{(x,y) \in S} P\nabla_\zeta L_\zeta(x, y, \theta')$$

$$G_S(P\zeta) = P\zeta - \gamma \sum_{(x,y) \in S} \nabla_{P\zeta} L_{P\zeta}(x, y, \theta').$$

Note that $\nabla_{P\zeta} L_{P\zeta}$ is not the same as $\nabla_\zeta L_{P\zeta}$, as the derivatives in the gradient vector of the former case shall match the parameter order in $P\zeta$ rather than $\zeta$. Since $\omega_P$ is a parameter space symmetry, by definition 2, $\forall \zeta \in (\mathcal{R}^m)^n$, $f_\theta$ and $f_{\omega_P(\theta)}$ are the same function, and thus $L_\zeta$ and $L_{P\zeta}$ are the same function. Therefore, it is straightforward that $\nabla_{P\zeta} L_{P\zeta}(x, y, \theta') = P\nabla_\zeta L_\zeta(x, y, \theta')$ for all $x, y, \zeta, \theta'$. This directly leads to $PG_S(\zeta) = G_S(P\zeta)$. $\square$

**Theorem 3** (Exchangeable Values). *Under the conditions of Theorem 2, for any NN input $x \in \mathcal{X}$, define $\xi_i = g_{\theta', \zeta_i}(x)$, where $g$ is the function of a sub-network in the trained model, parameterized by one $\zeta_i$ and possibly $\theta'$. We have that $\xi_i$'s have exchangeable distributions in the trained model.*

*Proof.* Since $\theta'$ is shared among $\xi_i$'s, the only variable in $\xi_i$ is $\zeta_i$. $(\zeta_1, ..., \zeta_n) \to (\xi_1, ..., \xi_n)$ is an exchangeability preserving transformation by Theorem 1. Therefore, $\xi_i$'s are exchangeable. $\square$

Note that given a group of exchangeable parameters $\zeta_i$'s, there might be multiple groups of exchangeable values $\xi_i$'s in NN computation, each with a different sub-network $g$.

---

[1]In fact, one can prove that permuting $\zeta_i$'s does not affect the update to other parameters in $\theta'$.

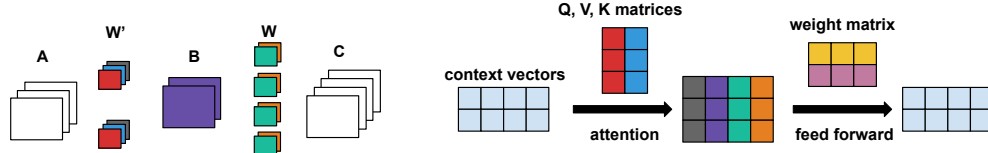

(a) Exchangeable parameters in CNNs        (b) Exchangeable parameters in transformers

Figure 4: Parameters with the same color have exchangeable distributions in the trained model.

$\xi' = <>$
**for** $i$ **in range**$(n)$ **do**
    Compute $\xi_i$, $\xi' = \xi' \oplus \xi_i$
    **if** confident$(\xi')$ **then**
        **return** $\rho(\xi')$
**return** $\rho(\xi)$

(a) EXPRUNE general algorithm.

$\xi' = <>$
**for** $i$ **in range**$(n)$ **do**
    Compute $\xi_i$, $\xi' = \xi' \oplus \xi_i$
    **if** pred$(\xi', w, b)$ **then**
        **return** 0
**return** ReLU$(w \sum_{i=1}^{n} \xi_i + b)$

(b) ReLU instantiation

$\xi' = <>$, scores $= <0,0,...,0>$
**for** $i$ **in range**$(n)$ **do**
    Compute $\xi_i$, $\xi' = \xi' \oplus \xi_i$
    scores $+= \xi_i$
    **if** dom$(\xi',$scores$)$ **then**
        **return** cur_winner
**return** argmax$($scores$)$

(c) top-1 prediction head instantiation

Algorithm 1: Pseudocode for EXPRUNE algorithms. $\xi_1,...,\xi_n$ are exchangeable values to be computed. $\rho$ is invariant to input permutations and can be approximately evaluated with fewer than $n$ input $\xi_i$'s.

## 3.3 EXCHANGEABLE PARAMETERS AND VALUES IN POPULAR NEURAL NETWORKS

Using the insights from Theorems 2 and 3, we identify exchangeable parameters and values in popular NN architectures. We omit the proof of parameter space symmetry when it is straightforward. The exchangeable parameters $\zeta_i$'s can often be found in a "*map-reduce*" pattern, where $\zeta_i$'s map the input into exchangeable values $\xi_i$'s, which are then reduced with a permutation-invariant function.

***MLP:*** Given two sequential fully-connected layers with weights and biases $(W', b')$ and $(W, b)$, we let $n$ be the number of neurons in the middle, $\zeta_i = W'_{i,:} \oplus b'_i \oplus W_{:,i}$ and $\xi_i = \sigma(W'_{i,:} a + b'_i) W_{:,i}$, or $\xi_i = \sigma(W'_{i,:} a + b'_i)$, where $\sigma$ is any activation function, and $a$ is any input. Since handling of biases in other NN structures is similar, we assume no bias in the following cases for simplicity.

***Convolutions:*** We analyze 2D convolution in CNNs with 1 group as an example (Figure 4a). The analysis easily extends to 1D or multi-dimensional convolutions and grouped convolutions, covering models such as GCNs. $A, B, C$ has $C_1, C_2, C_3$ channels respectively. We use $W_{i,j}$ (similarly for $W'$) to denote the kernel weights mapping the $j$-th input channel to the $i$-th output channel. Define the 2D convolution function conv, such that $B = \text{conv}(A, W')$. We first consider one structure in CNNs as follows. Refer to Appendix A.1 for the handling of other common structures in CNNs, normalization layers, and skip connections. When instantiating $\zeta$, the weight tensors are flattened when concatenated.

*Two consecutive convolution layers.* We instantiate $n = C_2$, $\zeta_i = W'_{i,:} \oplus W_{:,i}$, and $\xi_i = \text{conv}(\sigma(\text{conv}(A, W'_{i,:})), W_{:,i})$, or $\xi_i = \sigma(\text{conv}(A, W'_{i,:}))$, for any activation function $\sigma$.

***Transformers*** We analyze a decoder-only transformer architecture due to its popularity in recent LLMs (Touvron et al., 2023; Bai et al., 2023) (Figure 4b). We formalize single-head attention followed by a fully connected layer. This basic analysis can be extended to include normalization layers and skip connections similarly as in CNNs. Let $X$ be the input, $K, Q, V$ be the attention matrices, and $W$ be the weight of the fully connected layer. The function of these two layers is $WVX\sigma((QX)^T(KX))$, where $\sigma$ is column-wise softmax. We instantiate $n = d_2$ and $\zeta_i = V_{i,:} \oplus W_{:,i}$ (or $\zeta_i = K_{i,:} \oplus Q_{i,:} \oplus V_{i,:} \oplus W_{:,i}$). Permuting $\zeta_i$'s is a parameter space symmetry because $(WP^T)(PV)X\sigma((QX)^T(KX)) = WVX\sigma((QX)^T(KX))$, for any permutation matrix $P$. $\xi_i$ can be instantiated as the attention layer's output $\xi_i = V_{i,:}X\sigma((QX)^T(KX))$, or the final output $\xi_i = W_{:,i}V_{i,:}X\sigma((QX)^T(KX))$. Interestingly, this indicates that the intermediate activations in every decoder layer are exchangeable.

## 4 EXPRUNE ALGORITHM

We next present EXPRUNE algorithm, a dynamic pruning algorithm exploiting exchangeability. We first present the general EXPRUNE algorithm, and then two instantiations of it for specific structures.

Algorithm 1a presents the EXPRUNE general algorithm. Assume that we want to compute a sub-network with function $\rho(\xi)$, where $\xi_i$'s are exchangeable values. The input to the sub-network have been computed. $\rho$ is invariant to permutations of $\xi_i$'s, and can be approximated with $k$ ($k \leq n$) different $\xi_i$'s as input, where larger $k$ leads to more accurate approximation. The algorithm sequentially computes $\xi_i$'s, and prunes computation of the rest when certain statistics of the computed ones pass a confidence check. We emphasize the following points about the general algorithm.

- **Theoretically grounded**. The algorithm only works when $\xi_i$'s have exchangeable distributions, because otherwise the statistics of some $\xi_i$'s are not informative about the other $\xi_i$'s.
- **Generality**. The algorithm can be instantiated to process sub-networks of different granularities (e.g., neurons, channels, layers) with different $\rho$ function. We note that the algorithm usually achieves better results for $\rho$ functions with locally flat regions, as it tolerates small approximation errors in the input.
- **Computation order**. $\xi_i$ can be computed in any order with the same expected accuracy due to their exchangeability. Optionally, they can be computed in certain heuristic orders.
- **Checking frequency**. For simplicity, we present the algorithm as checking confidence after computing each $\xi_i$. In practice, to reduce overhead, EXPRUNE can use less checks, e.g., check after computing every 32 $\xi_i$'s, or only check once at $i=32$.
- **Composibility**. The algorithm works with unmodified models and training algorithms. Therefore, it can compose with techniques that optimize the model statically, e.g., quantization and static pruning.

## 4.1 INSTANTIATION 1: EARLY NEGATIVE PREDICTION FOR RELU

Algorithm 1b presents an instantiation of EXPRUNE that prunes computation of each neuron's activation when $\rho$ is the activation function ReLU. ReLU is widely present in various NN architectures across various application domains. Recent transformer models switch to other activation functions (e.g., GELU (Hendrycks & Gimpel, 2023) and SiLU (Elfwing et al., 2017)) for faster training convergence, but Mirzadeh et al. (2024) demonstrated that for efficient inference, ReLU can replace them after training, leading to negligible accuracy loss and up to 90% sparsity with lightweight finetuning. EXPRUNE exploits the property of ReLU that it outputs zero for negative input, and prunes the computation if the final sum is predicted to be likely negative. In some cases, the sum of $\xi_i$'s is not directly processed by ReLU, but is first scaled and added biases, e.g., normalization layers, layer biases, shortcut connections. They can all be taken into account by a scaling weight $w$ and a bias term $b$. We devise two negative prediction methods as follows (assuming $w=1, b=0$ for simplicity).

- **Threshold.** Check if the current mean is below a predetermined threshold $\sum_{i=1}^{k} \xi_i' < kT$. This simple method requires only one more comparison, as the running sum is computed by NN already.
- **StatsTest.** We perform a Wald's test (Wald, 1992) with confidence level $\alpha$, checking if

$$\Phi\Big(\frac{\frac{1}{n}\sum_{i=1}^{k}\xi_i'}{\sqrt{\frac{1}{n}\sum_{i=1}^{k}\xi_i'^2 - (\frac{1}{n}\sum_{i=1}^{k}\xi_i')^2}}\Big) < \alpha, \text{ or equivalently } \frac{(\sum_{i=1}^{k}\xi_i')^2}{k\sum_{i=1}^{k}\xi_i'^2 - (\sum_{i=1}^{k}\xi_i')^2} < (\Phi^{-1}(\alpha))^2$$

where $\Phi$ is the cumulative density function of the standard normal distribution, and the right hand side of the simplified inequality can be stored as constant. This method introduces overhead that scales linearly the number of $\xi_i$'s, as it requires computing running sum of the squared term over the $k$ terms in the partial results. Note that the assumptions of Wald's test are not strictly met by all exchangeable sequences. We discuss the assumptions in more detail in Section 6.

## 4.2 INSTANTIATION 2: DOMINANCE PREDICTION FOR PREDICTION HEADS

Algorithm 1c presents an instantiation of EXPRUNE for prediction heads, in which each $\xi_i$ accumulates scores to every class, and the class with the maximum score is returned. This structure is widely present in NNs for classification and retrieval tasks. The amount of compute incurred by the prediction heads is a small portion in large models, but non-trivial in edge models. The algorithm predicts whether the current winner class (with the maximum score) likely dominates the others if all $\xi_i$'s are computed. We provide two possible dominance prediction methods $\mathrm{dom}$, similar to Section 4.1.

- **Threshold**. Prune if a set of conditions $c_1 - c_i > T_i$ are met, where $c_i$ is the $i$-th largest score, and $T_i$'s are predetermined thresholds.
- **StatsTest**. We can conduct a Wald's test for the score of $\mathrm{cur\_winner}$ against each other class, and prune when all or a subset of the tests pass. Since multiple tests are involved, Holm-Bonferroni method (Holm, 1979) can be used to assign adjusted confidence levels for tests, given the overall $\alpha$.

Table 1: Datasets and models. BN means the model is enhanced with BatchNorm Ioffe & Szegedy (2015). The default fidelity metric is ROC-AUC for ogbg-molhiv and accuracy for others. GCN is graph convolutional NN. $^\dagger$ indicates the EXPRUNE baseline also optimizes the model's prediction head.

| Task | Dataset | Models (# Parameters) |
|---|---|---|
| Image Classification$^\dagger$ | CIFAR100 Krizhevsky (2009) | ResNet18-BN He et al. (2015) (22.4M) |
| Graph Property Prediction | ogbg-molhiv Wu et al. (2018) | GCN Kipf & Welling (2017) (527K) |
| Question Answering | PIQA Bisk et al. (2020) | OPT Zhang et al. (2022) (6.7B) |

## 5 EVALUATION

We compare EXPRUNE against the unoptimized inference and similar prior work on various models with ReLU activation functions. Table 1 summarizes our benchmarks and models.

***Datasets and Models.*** We have 3 datasets, covering 3 different tasks and input types. Please refer to the Appendix A for the details of dataset split and obtaining the trained model weights. We choose OPT (Zhang et al., 2022) because it is an off-the-shelf ReLU activated language model. Mirzadeh et al. (2024) demonstrated that one can replace ReLU in LLMs with other activation functions without accuracy loss, but they did not release the weights of their "reluficated" models.

***Metrics.*** We use the default fidelity metrics for each dataset. We use the floating-point operation performed (FLOPs) as our performance metric, as it is a good proxy for inference efficiency (Mirzadeh et al., 2024). We count the FLOPs incurred by dynamic pruning algorithm towards the total FLOPs.

**EXPRUNE *Baselines.*** EXPRUNE baselines use ReLU instantiation for all models on all applicable sub-networks, and prediction head instantiation for ResNet18-BN model (the other models do not have the top-1 prediction head), to prune different parts of the model. The prediction head instantiation does one STATSTEST after processing $k = 160$ terms $\xi_i$'s, testing the score of the current top-1 against top-2 and top-3, with overall $\alpha = 0.1$. The ReLU instantiation can use either prediction metric described in Section 4.1. We try both and have two EXPRUNE baselines dubbed THRESHOLD and STATSTEST. For ReLU layers, EXPRUNE performs one negative prediction after processing $k = 32$ terms $\xi_i$'s and terminate if confident, otherwise computes all other $\xi_i$'s. We choose the number 32 as many statistics methods target sample sizes of at least 30 (VanVoorhis et al., 2007). THRESHOLD uses one FLOP as the threshold $kT$ is stored as a constant. STATSTEST uses $2k + 6$ FLOPs as it uses $2k$ FLOPs for computing the sum of $\xi_i'^2$, and 6 more to compute the Wald's statistic and compare it to the stored threshold. We report FLOPs of the whole model for CNNs and GCNs as EXPRUNE is applied to most of the computation, and the FLOPs of all the linear layers in between an attention layer and a ReLU activation for OPT, as EXPRUNE is only applied to these layers. The FLOPs of these linear layers account for approximately $1/3$ of the total FLOPs in the unoptimized inference (Ding et al., 2024; Zhang et al., 2022).

***Hyperparameter Optimization.*** As different layers in NN may have different error sensitivities, we use Optuna (Akiba et al., 2019) to find optimal parameter combinations for EXPRUNE. Specifically, each layer has one parameter $T$ (THRESHOLD) or $\alpha$ (STATSTEST). We tune the hyperparameters for 2000 trials on the validation set. Refer to Appendix A for Optuna configuration details. Using the data we have on validation set, we select a subset of promising parameter configurations to run on the test set. This emulates the process of selecting the hyperparameter combination for deployment. We iteratively select all the combinations on the fidelity-FLOPs Parato Frontier (dubbed one Parato slice), remove them from the set, and choose all on the next Parato slice. We include all points on the first 5 Parato slices.

***SnaPEA baseline.*** The prior work most similar to EXPRUNE is SnaPEA (Akhlaghi et al., 2018), as it also targets neuron-level pruning with ReLU activations. SnaPEA does error-free dynamic pruning by sorting the weights offline so that the partial sum starts monotonically decreasing after processing some terms, and then pruning the computation when the sum drops below zero. However, it requires that every layer's output is non-negative, and there is no normalization layers and shortcut connections, thus unable to directly apply to any model in our evaluation. To make our best attempt to compare with it, we make it work for ResNet18-BN by changing the model architecture, fusing the batch normalization into the convolution weights, and then adapting the SnaPEA algorithm to take into account the shortcut connection. We note that similar adaptations cannot make SnaPEA work for other models in our evaluation set.

### 5.1 MAIN RESULTS AND ANALYSIS

Figure 5 shows the performance of the baselines. We use the default evaluation order of $\xi_i$'s. Across three models and compared to the unoptimized baseline, EXPRUNE is able to deliver 10.98–17.33% re-

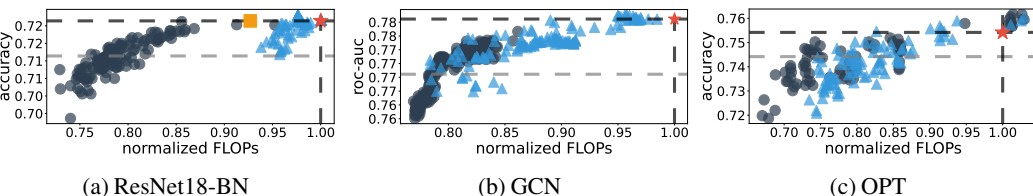

Figure 5: ● is STATSTEST, ▲ is THRESHOLD, ■ is SNAPEA, ★ is the unoptimized baseline. — show fidelty and normalized FLOPs for unoptimized baseline. — shows baseline fidelity minus 1%.

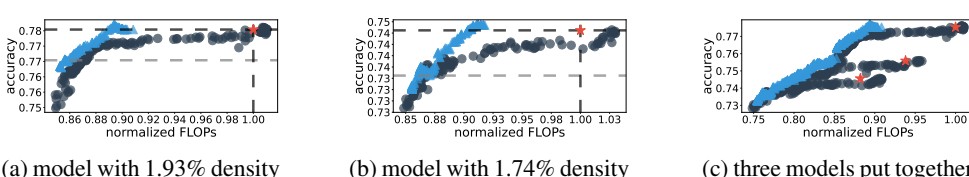

(a) model with 1.93% density     (b) model with 1.74% density     (c) three models put together

Figure 6: Fidelity-FLOPs scatter plots for statically pruned VGG11-BN models. FLOPs are normalized to largest model's unoptimized baseline in (c). Colors and lines have the same meaning as in Figure 5.

duction in FLOPs with negligible (<0.1%) fidelity drop, and 21.61–27.16% reduction in FLOPs with at most 1% fidelity drop. On ResNet18-BN, while SnaPEA delivers only 7.3% FLOPs reduction without accuracy loss with on average 18.5 checks per neuron, EXPRUNE delivers 14.4% FLOPs reduction with only one check per neuron, introducing larger FLOPs reduction with less branching, making hardware acceleration easier. We find that STATSTEST performs much better than THRESHOLD for CNNs, because STATSTEST offers more accurate negative prediction. In GCN and OPT, STATSTEST and THRESHOLD perform similarly. This is because it takes fewer FLOPs to compute each exchangeable value $\xi_i$ in the 1D convolution of GCN and the linear layer of OPT, compared to the 2D convolution in CNNs, and thus the overhead of STATSTEST acounts for a larger portion in the total FLOPs.

## 5.2   COMBINING EXPRUNE WITH STATIC MAGNITUDE PRUNING

We demonstrate that EXPRUNE can be applied to statically pruned models and offer additional reduction in FLOPs. We train a VGG11-BN model (Simonyan & Zisserman, 2015) on CIFAR10, and iteratively (1) set the 5% parameters in all convolution kernels with smallest magnitude to zero, and (2) finetune the model on the training set to recover accuracy, following the practice of Han et al. (2015). Note that exchangeability is also present in the pruned model, as exchangeable parameters have the same probability to be pruned in the process. We take the models at three consecutive pruning iterations, where only 1.93%, 1.83%, and 1.74% weights in convolution layers are left. We choose aggressively pruned models because we want to study how EXPRUNE works with already extremely compressed models, in which static pruning cannot compress more without accuracy loss. To work with these models, we let EXPRUNE compute $\xi_i$'s in the order of computation cost, i.e., the number of none-zero weights in the corresponding $\zeta_i$. In other words, we compute the cheap $\xi_i$'s first. This corresponds to sorting channels of convolution kernels, which can be done statically offline and introduces no overhead during inference.

The results are shown in Figure 6. In each of the three pruned models, EXPRUNE still provides 10.24–11.11% reduction in FLOPs with negligible accuracy drop, and 13.91–14.39% reduction in FLOPs with at most 1% accuracy drop, compared to the unoptimized inference. THRESHOLD achieves better results than STATSTEST in pruned models, because the exchangeable values $\xi_i$'s are cheaper to compute in these models, making the overhead of STATSTEST offset the FLOPs reduction of computing fewer $\xi_i$'s. Figure 6c shows all the points in three models compared together. We find that EXPRUNE combined with static pruning achieve better accuracy-performance trade-off than only static pruning. This indicates EXPRUNE composes with static pruning, because EXPRUNE can remove redundancy that cannot be removed by static pruning.

## 6   DISCUSSION

***Exchangeability and Confidence Test.*** De Finetti's theorem states that infinite sequence of exchangeable random variables are conditionally $iid$. Exchangeability of a finite sequence of random variables could indicate either that they are conditionally $iid$, or that they form a case of sampling without replacement. The latter case does not satisfy the assumptions made by Wald's test (Wald,

1992), but it can be approximated with an $iid$ (Diaconis & Freedman, 1980). It is interesting and valuable to investigate stronger statistical properties to describe them. It is of great practical value to devise better confidence test for EXPRUNE that is more accurate and more efficient.

**EXPRUNE *and Hardware Acceleration.*** EXPRUNE reduces FLOPs but also breaks certain structures of computation which are exploited in hardware accelerators, e.g., uniformity for parallel processing. We note that in resource-constrained scenarios such as edge/embedded ML with limited parallelism, FLOPs reduction straightforwardly translates to speed-up and energy savings. Customized architecture/hardware such as the one proposed with SnaPEA (Akhlaghi et al., 2018) also helps translate FLOPs reduction of dynamic pruning into speed-up and energy reduction. EXPRUNE could also be integrated with scheduling algorithms of reconfigurable dataflow architectures such as CGRAs (Koul et al., 2023) to reduce the amount of energy-intensive off-chip memory loading.

## 7 RELATED WORK

***Exchangeability in Deep Learning.*** Statistical exchangeability has various applications in deep learning, but prior work focused on exchangeable data. Observing exchangeability of certain input data such as sets, special NN architectures have been proposed to process them (Chan et al., 2018; Korshunova et al., 2018; Bloem-Reddy & Teh, 2020; Wiese et al., 2023). Conformal prediction provides a prediction set with guaranteed error rate for any NN model assuming exchangeability of data sequence (Fontana et al., 2023; Kuchibhotla, 2020).

***Symmetry in Deep Learning.*** Symmetry in NNs and its impact have been extensively studied by theoreticians. Symmetry is known to affect model generalization (Dinh et al., 2017), interpretability (Godfrey et al., 2022), and the loss landscape (Zhao et al., 2023; Lim et al., 2024). A related but different concept is equivariance (Zaheer et al., 2017; Cohen & Welling, 2016), a feature of special NN architectures that the NNs produce consistent outputs under symmetry transformations of the inputs.

***Static NN Model Optimizations.*** Various techniques have been proposed to derive efficient NN models with smaller sizes, including pruning (Han et al., 2015; Cheng et al., 2024), quantization (Saha et al., 2024; Gholami et al., 2022; Hubara et al., 2016; Qin et al., 2020), knowledge distillation (Hinton et al., 2015; Gou et al., 2021), and neural architectural search (Elsken et al., 2019). These methods are statically applied before model deployment. In contrast, ours is dynamic and applies on a per-input basis during inference. In our evaluation (Section 5.1), we show that our method composes with static pruning.

***Dynamic Pruning at Inference.*** Researchers have explored coarse-grained dynamic pruning methods, which are applied on a per-input basis during NN inference (Cheng et al., 2024). They explored dynamically pruning layers (Teerapittayanon et al., 2016; Tambe et al., 2021; Han et al., 2022), tokens (Anagnostidis et al., 2023), channels (Lin et al., 2017; Elkerdawy et al., 2022), and spatial domain (Liu et al., 2018). These methods are specialized to certain model architectures and coarse-grained, operating on structures larger than neurons. In contrast, our method can operate at very fine granularity (neuron level) and can be generalized across multiple architectures and granularities. Our method can also potentially compose with these approaches as we work at different network granularity.

SnaPEA (Akhlaghi et al., 2018) and ComPreEND (Kim et al., 2022) explored neuron-level dynamic pruning for convolution. They make similar assumptions that do not hold for modern model architectures (discussed in Section 5). ComPreEnd is more limited, requiring specific architecures and fixed-point number representations. Another line of work (Wakatsuki et al., 2021; Kong et al., 2023) exploited similar patches in input feature maps to derive upper bound of the weighted sum given the partial sum, and terminated early when the upper bound is below zero. These methods only take effect on similar input patches and are specialized to CNNs and video processing. In contrast, our method works with many model architectures, and does not require similar input patches.

## 8 CONCLUSION

We present a novel theory that formalizes exchangeability between certain parameters and intermediate values. We identify exchangeable parameters and intermediate values in popular NNs. Exploiting this insight, we devise a general dynamic pruning algorithm EXPRUNE using statistics of partial evaluation results. We present two instantiations of EXPRUNE for ReLU activations and prediction heads respectively. We demonstrate that EXPRUNE is able to provide large FLOPs reduction in image CNNs, GCNs, and LMs. We also show that EXPRUNE is able to compose with static pruning, providing additional FLOPs reduction on models that are heavily pruned statically.

## REPRODUCIBILITY STATEMENT

Our evaluation details are described in the Appendix A. Our evaluation code is released at `https://anonymous.4open.science/r/Exchangeable-NN-FBC7`.

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

## A  Technical Appendices and Supplementary Material

### A.1  Exchangeable Parameters and Values in CNNs and Embeddings

**Convolutions.**  We present the CNN formulation and the exchangeability in two consecutive convolutions in Section 3.3. Here we present the exchangeability in other common structures in CNNs.

*Normalization Layers.* Normalization layers (NL) such as batch normalization Ioffe & Szegedy (2015) and layer normalization Ba et al. (2016) can stabilize training. It applies a channel-wise scaling and bias. In CNNs, one normalization layer is usually inserted right before or after each activation function. We present an analysis for the latter case but the analysis also applies to the former case. Denote the NL functions as NL with parameters nl, and $\text{NL}_i$ denotes the NL applied to the $i$-th channel, parameterized by $\text{nl}_i$. We instantiate $n = C_2$, $\zeta_i = W'_{i\cdot} \oplus \text{nl}_i \oplus W_{\cdot i}$, and $\xi_i = \text{conv}(\text{NL}_i(\sigma(\text{conv}(A, W'_{i\cdot}))), W_{\cdot i})$.

*One convolution layer followed by a fully connected layer.* For simplicity, we assume $B$ goes through channel-wise average pooling $\text{pool}$ ($\text{pool}(B)$ is of shape $C_2$) before the fully connected layer. The analysis easily generalizes to other/no pooling as well. The second layer then has function $W\,\text{pool}(B)$. We instantiate $n = C_2$, $\zeta_i = W'_{i\cdot} \oplus W_{\cdot i}$, and $\xi_i = \text{pool}(\text{conv}(A, W'_{i\cdot}))W_{\cdot i}$.

*Skip Connections.* Skip connections from $A$ to $C$ do not affect the rest of the analysis. Skip connections from a layer before $A$ to $B$, and from $B$ to a layer after $C$ are similar, and we present an analysis for the former case. We can simply include in $\zeta_i$ the parameters that produce, and also the parameters that consume, the $i$-th channel of the shortcut values. The rest of the analysis is unaffected.

**Embeddings** Embedding dimensions are intuitively symmetric because embeddings are "distributed" representations as the relevant information is represented in many dimensions. We present the simple example Word2vec Mikolov et al. (2013). Let $m$ be the number of words, $n$ be the embedding dimension, $A$ be the embedding matrix for all the words, $M$ be the fully connected layer weight matrix (both of size $m \times n$), and $\sigma$ be the softmax operation. The NN takes into input a word index $k$, and outputs $\sigma(MA_k^T)$. We instantiate $\zeta_i = A_{\cdot i} \oplus M_{\cdot i}$ and $\xi_i = A_{ki}M_{\cdot i}$. Alternatively, $\xi_i = A_{ki}$, which indicates that learned embedding dimensions are exchangeable.

### A.2  Evaluation Details

All the details can be found in our codebase at `https://anonymous.4open.science/r/Exchangeable-NN-FBC7`.

**Dataset split.** The test set labels of PIQA Bisk et al. (2020) are not published, so we use the validation set as the test set, and 10% of the training set as the validation set. This is reasonable because the pretrained OPT model was not trained on PIQA training set. For CIFAR100 Krizhevsky (2009), we use a fixed split of the training set, with 90% samples used for training, 10% used as validation set, and use the default test set. We use the default dataset splits for ogbg-molhiv dataset Wu et al. (2018).

**Model architectures.** We use the default OPT architecture from HuggingFace Wolf et al. (2020). The GCN architectures follow Kipf and Welling Kipf & Welling (2017). We use the adapted CNNs He et al. (2015) for CIFAR100. Specifically, the first layer and the pooling layer before the prediction head are adapted in size for the image size and class number in CIFAR100.

**Obtaining trained models.** We train image CNNs and GCNs locally using the training set, and used the pretrained weights for OPT Zhang et al. (2022) from HuggingFace Wolf et al. (2020). For local training of ResNet18-BN, we use AdamW optimizer, $\times 10^{-2}$ learning rate, 16 batch size, $10^{-4}$ weight decay, 1cycle learning rate scheduler, and 100 epochs. When statically pruning VGG11-BN on CIFAR10, we use the same training scheme in every pruning iteration to finetune the model after 5% parameters in all convolution kernels are set to zero. For local training of GCN, we use AdamW optimizer, $10^{-3}$ learning rate, 32 batch size, $10^{-4}$ weight decay, 1cycle learning rate scheduler, and 80 epochs.

**Optuna hyperparameter tuning.** Across all models, we set the range of $\alpha$ as $[0, 0.5]$ in STATSTEST for all layers. The range of $T$ in THRESHOLD is $[-30, 0]$ for CNNs, $[-1, 0]$ for GCN, $[-0.005, 0]$ for OPT. For statically pruned models, we additionally add a parameter $r$ in range $[0.1, 0.5]$ to tune for each layer, which controls when EXPRUNE is disabled. When the ratio of the total FLOPs that can be potentially pruned for a channel's computation to the overhead of EXPRUNE is below $r$, EXPRUNE is disabled. For each model, we provide a set of initial points to warm up Optuna's surrogate model. Please see our code for details.