# OpenReview forum: "Exchangeability in Neural Networks and its Application to Dynamic Pruning"
_ICLR.cc/2026/Conference — Submitted to ICLR 2026_

### Official Review · Reviewer_HzWb · 2025-10-31

**Soundness:** 2
**Presentation:** 3
**Contribution:** 2
**Rating:** 2
**Confidence:** 4

**Summary:**

This paper introduces EXPRUNE, a novel dynamic pruning method grounded in the statistical concept of exchangeability. The authors formalize how certain neural network parameters and intermediate values possess exchangeable distributions across trained models (viewed as samples from a distribution over random initializations). By identifying exchangeable parameter groups, EXPRUNE enables partial computation followed by statistical confidence checks to prune remaining operations on a per-input basis.

**Strengths:**

1. The term "Exchangeable parameters" looks novel in the area of pruning.

2. Both CNNs and transformers are considered.

**Weaknesses:**

1. Some statements are not clear. An example is Figure 4. What's the meaning of different colors? Why does activation B only have a common color with weight W', but not W?

2.  Limited experiments.

2.1 No ImageNet experiments, which is the standard benchmark for evaluating CNN pruning methods

2.2 Lack of comparison with recent strong baselines in structured pruning and dynamic inference

2.3 No wall-clock time measurements. The paper only reports FLOPs, which is an imperfect proxy for actual speedup. Section 6 (line 435-442) acknowledges hardware acceleration challenges but provides no real timing data. Without wall-clock measurements, it's impossible to assess practical value

3 Prohibitive hyperparameter tuning cost.

3.1 Line 361: 2000 Optuna trials per model on the validation set represents enormous computational cost.

3.2 Figure 3 (line 072) requires training **500 models** from different random initializations to demonstrate exchangeability. This massive training cost is not accounted for when claiming "efficiency" gains. In practice, users have only **ONE trained model**, making the theoretical exchangeability properties impossible to verify

4. Line 749 specifies vastly different threshold ranges for different architectures: [-30,0] for CNNs, [-1,0] for GCN, [-0.005,0] for OPT. These ranges appear to be manually defined without theoretical guarantees or principled derivation

5. So many other dynamic pruning methods exist (layer-wise early exit, token pruning for transformers), but are not compared.

6. The code link at line 488 does not work properly. Except for the README file, all other files show “The requested file is not found.”

**Questions:**

See weaknesses

---

> ### Author Response · Authors · 2025-11-21
> **Response**
>
> # Writing not clear
>
> Could you elaborate on which statements you found confusing?
>
> For Figure 4, we acknowledge that the meaning of different colors were not explained in detail. In the image caption, we explained that “Parameters with the same color have exchangeable distributions in the trained model”. For example, in Figure 4 (a), the first channels of both convolution kernels in W’ (denoted as W’[0][0] and W’[1][0] respectively) are red. This means that for any (i,j), the two corresponding weights W’[0][0][i][j] and W’[1][0][i][j] have exchangeable distributions with respect to random initializations. Similarly, in Q, V, or K matrices, element (i,j) and element (i’,j) have the same color, as they also have exchangeable distributions. Different colors indicate that the parameters may not have exchangeable distributions.
>
> > Why does activation B only have a common color with weight W', but not W?
>
> You might have misread the grey/blue color in W’ as purple. The activation B does not have a common color with W’ or W.
>
> # Cost of hyperparameter tuning
>
> Thanks for the question. We note that the hyperparameter tuning would only be done once before deployment, and the cost of tuning is actually low. We note that although the 2000-trial optimization might look a lot, the actual cost is very small and negligible compared to the training cost, as each trial involves only one forward pass on the validation set. For CNNs and GNNs, the validation set is only ~10% the size of the training set. For LLMs, the validation set consists of only 2000 question-answering pairs, which contain much fewer tokens than the training text. Each of the 2000 trials only does a forward pass. Considering that the back propagation is more costly than the forward pass, the tuning is less costly than a 100-epoch training. We note that training also involves hyperparameter tuning and requires multiple reruns and each run usually takes more than 100 epochs. Compared to the training cost and the savings on a lot of inference runs, the cost of ExPrune hyperparameter tuning is negligible.
>
> # Wall-clock time measurements
>
> Thanks for your suggestion. We agree that wall-clock latency or memory-access numbers could provide more insights into the actual performance benefits of ExPrune. We have ongoing collaboration with hardware/architecture researchers that deploys the ExPrune algorithm in dataflow architectures, and our results indicate that it can provide ~30% reduction in memory access and more than 2x end-to-end speedup.
>
> # Comparison with recent strong baselines in structured pruning and dynamic inference
>
> We would like to note that ExPrune is training free and calibration free, does not require architectural changes to the model, and is general across model architectures. This makes many pruning methods not comparable. For example, the layer-wise early exit requires architectural changes to the model and is not training free. The token pruning is specific to transformers. Furthermore, ExPrune can potentially compose with these methods.
>
> # ImageNet experiments
>
> We acknowledge that it would be better to evaluate the CNNs on the ImageNet dataset. We are currently working on scaling the methods to ImageNet.
>
> # Figure 3, 500 models
>
> The users do not need to train 500 models to verify the theoretical exchangeability properties — we have mathematically proved them. The reason we got 500 models is to demonstrate that our theory holds for pedagogical reasons. The users can find out exchangeable parameters and values with pen and paper following our theory.
>
> # Different threshold ranges for different architectures
>
> We define different threshold ranges for different architectures because different architectures have different activation ranges due to normalization layers, etc. We set the ranges according to the activation statistics collected from the model on a small number of inputs. We note that we could also set a large range and let Optuna narrow the range, but it could cost Optuna more trials.
>
> # The code link at line 488
>
> Could you check again? I can access all the files even from an incognito page. It might be due to the issues with the Anonymous GitHub.

---

> > ### Comment · Reviewer_HzWb · 2025-11-26
> >
> > Thanks for the reply.
> >
> > >  Considering that the back propagation is more costly than the forward pass, the tuning is less costly than a 100-epoch training.
> >
> > Can you provide the detailed time required for training and tuning?
> >
> > The paper claims to be “training-free”, yet the method still performs extensive tuning, which is not cost-negligible. To fairly support the comparison between “training-free + tuning” (your method) and “re-training + no tuning” (prior work), please provide concrete computational numbers:
> >
> > - Prior work: total GPU-hours / time required for full re-training.
> >
> > - Your method: total GPU-hours / time required for the tuning stage.
> >
> > A quantitative comparison is necessary.
> >
> >
> > Apart from this, I'm waiting for more details, such as wall-clock time measurements and ImageNet experiments.

---

> > > ### Author Response · Authors · 2025-12-03
> > >
> > > # The detailed time required for training and tuning
> > >
> > > We first want to note that the benefit of training-free methods, besides reduced cost, is that they are compatible with any training/finetuning pipelines/tricks/approaches, and they do not permanently reduce the model capacity by removing parameters that may be important. Below we provide a runtime comparison for the ResNet-18 model on our MacBook with a M3 chip:
> > >
> > > Training hyperparameter tuning (with heavily optimized training kernels): Each 100-epoch training run takes ~2.3h. Practitioners usually do a sweep of hyperparameters, consisting of tens to hundreds of training runs (e.g., 50 [1]). That totals to hundreds if not thousands of hours. Even if only accounting for the retraining cost of the pruning methods in prior work, they usually also use five to tens of training runs [2, 3], totaling at least ~11h if run on our laptop.
> > >
> > > ExPrune hyperparameter tuning (without any specialized optimization from us): Each Optuna trial takes on average ~12s on validation set. Even if we do 2000 trials, that only uses a total of ~6.6h.
> > >
> > > # Wall-clock time measurements and ImageNet experiments
> > >
> > > We want to note that we are the first to model the relationship among NN parameters with exchangeability and to perform partial computation during inference exploiting the property. We have demonstrated that the ExPrune algorithm generally works for 3 model families across 3 problem domains, with the largest model size containing 7.6B parameters, and the current paper is already packed with theory, algorithm, and a comprehensive evaluation. Scaling the algorithm to even larger models and larger datasets could be another interesting problem to be solved in another academic paper. To demonstrate the wall-clock time even in simulation, one needs to design a suitable compute architecture for the algorithm, which in itself could be a sufficient contribution for another academic paper.
> > >
> > > [1] Morvan, Marine Le, and Gaël Varoquaux. "Imputation for prediction: beware of diminishing returns." The Thirteenth International Conference on Learning Representations (2025).
> > >
> > > [2] Nowak, Aleksandra, Bram Grooten, Decebal Constantin Mocanu, and Jacek Tabor. "Fantastic weights and how to find them: Where to prune in dynamic sparse training." Advances in Neural Information Processing Systems 36 (2023): 55160-55192.
> > >
> > > [3] Han, Song, Jeff Pool, John Tran, and William Dally. "Learning both weights and connections for efficient neural network." Advances in neural information processing systems 28 (2015).

---

### Official Review · Reviewer_3JLL · 2025-11-03

**Soundness:** 3
**Presentation:** 3
**Contribution:** 3
**Rating:** 6
**Confidence:** 3

**Summary:**

This paper presents EXPRUNE, a novel, general dynamic pruning optimization designed to reduce the computational cost of neural network inference on a per-input basis. The method is grounded in the authors' theoretical finding that certain model parameters and intermediate values exhibit a statistical property called exchangeability, which implies they are identically distributed and have symmetric interdependence. EXPRUNE leverages this by partially evaluating a network computation, analyzing the statistics of the partial result, and making an on-the-fly decision to prune the rest of the computation if a confident prediction can be made (e.g., the final sum will be negative, resulting in a zero output from ReLU). A key advantage is that EXPRUNE requires no changes to the model architecture or training algorithm, allowing it to generalize across vision, graph, and language models. The evaluation shows it can reduce FLOPs by 10.98-17.33% with negligible accuracy drop and also composes with static pruning to provide additional performance gains.

**Strengths:**

1. For me, the paper is the first to model the relationship between neural network parameters and intermediate values using the statistical concept of exchangeability. The theoretical insight enables a novel dynamic pruning mechanism by using partial computation as a statistical sample to predict the outcome of the full computation.

2. The method is tested on three distinct and important domains: computer vision (ResNet18-BN), graph property prediction (GCN), and language (OPT), demonstrating the generality of the approach.

3. The paper is well-written and structured logically, making it easy to follow.

**Weaknesses:**

1. The central performance metric is FLOPs, which is acknowledged as a "proxy for inference efficiency". However, this is a weak proxy for a dynamic pruning algorithm. A stronger evaluation would include wall-clock time or memory-access metrics on target hardware.

2. The paper claims EXPRUNE is a "general, dynamic pruning optimization" that "generalizes across model architectures". While the inclusion of CNNs, GNNs, and a Transformer is good, the bulk of the evaluation relies on the ReLU activation function. Can the core idea be adapted to modern activations like GELU/SiLU?

3. The method looks sensitive to layer-specific hyperparameters. These are found using a 2000-trial optimization search with Optuna, which is a very costly and complex tuning process.

Minor:

1. Some notations are reused. For example, $b$ represents input/output in Section 2 and the model's bias parameter in Section 3.

**Questions:**

Why select the Wald's test, given that “the assumptions of Wald's test are not strictly met by all exchangeable sequences”?

---

> ### Author Response · Authors · 2025-11-21
> **Response**
>
> # FLOPs as a proxy for inference efficiency
>
> Thanks for your suggestion. We agree that wall-clock latency or memory-access numbers could provide more insights into the actual performance benefits of ExPrune. We have ongoing collaboration with hardware/architecture researchers that deploys the ExPrune algorithm in dataflow architectures, and our results indicate that it can provide ~30% reduction in memory access and more than 2x end-to-end speedup.
>
> # Can the idea of ExPrune be adapted to GELU/SiLU?
>
> We want to note that while GELU and SiLU are known to lead to faster convergence in LLM training, for efficient inference, a recent trend is to reinstate ReLU activations after training, because ReLU introduces much more sparsity. A recent work (ICLR 2024 Oral, cite [42] in the paper) found out that reinstating ReLU in various LLMs with a bit of fine-tuning leads to no accuracy degradation and up to 90% sparsity. In practice, one can use GELU/SiLU to train the LLMs, and switch to ReLU to use ExPrune. As discussed in section 4 of the paper, it is more straightforward to apply ExPrune for functions with locally flat regions, as they tolerate small approximation errors. It is less straightforward to use ExPrune for continuous functions such as GELU/SiLU, and the confidence metric presented in the paper can not be easily extended to work effectively for GELU/SiLU.
>
> Naively, one can use the instantiated ExPrune by setting a very low termination threshold, e.g., -4, as GELU/SiLU is very close to 0 when the input is smaller than 4. This will not degrade accuracy, but will also not reduce the FLOPs by much, since the input values are often above -4. As an interesting future work direction, one could try to design confidence metrics for GELU/SiLU, e.g., determining whether to terminate based on other statistics such as the sample variance of the computed exchangeable values.
>
> # Cost of hyperparameter tuning
>
> Thanks for the question. We note that the hyperparameter tuning would only be done once before deployment, and the cost of tuning is actually low. We note that although the 2000-trial optimization might look a lot, the actual cost is very small and negligible compared to the training cost, as each trial involves only one forward pass on the validation set. For CNNs and GNNs, the validation set is only ~10% the size of the training set. For LLMs, the validation set consists of only 2000 question-answering pairs, which contain much fewer tokens than the training text. Each of the 2000 trials only does a forward pass. Considering that the back propagation is more costly than the forward pass, the tuning is less costly than a 100-epoch training. We note that training also involves hyperparameter tuning and requires multiple reruns and each run usually takes more than 100 epochs. Compared to the training cost and the savings on a lot of inference runs, the cost of ExPrune hyperparameter tuning is negligible.
>
> # Why select the Wald's test, given that “the assumptions of Wald's test are not strictly met by all exchangeable sequences”?
>
> Thanks for the insightful question. The reasons are twofold. First, the statistical tests that apply to all exchangeable sequences are usually weaker than Wald’s test, e.g., non-parametric rank tests. They give weaker conclusions that are hard to translate to non-trivial reduction of computation. On the other hand, although Wald’s test does not strictly apply to all exchangeable sequences, it achieves good empirical results. Though it is hard to prove that the central limit theorem (the assumption of Wald’s test) holds, it usually empirically holds for exchangeable sequences, and when it does, the false positive rate for our negative prediction would be upper-bounded by the $\alpha$ parameter. Second, from the performance perspective, the Wald’s test introduces very small overhead, as it only requires computing a running sum of the squared terms for estimating the sample variance. The overhead is easily offset by the additional accuracy for predicting negativity provided by the Wald’s test.

---

> > ### Comment · Reviewer_3JLL · 2025-11-28
> >
> > Thanks for the responses, which address some of my questions.  I appreciate the insightful analysis of the "exchangeability" in NNs. The skepticism about the practical versatility remains (restricted to ReLU activation).   So I keep my score.

---

### Official Review · Reviewer_KE6v · 2025-11-05

**Soundness:** 2
**Presentation:** 2
**Contribution:** 2
**Rating:** 0
**Confidence:** 4

**Summary:**

The article proposes a dynamic pruning algorithm. The algorithm stands on a key concept called "exchangeability" that checks the presence of exchangeable model parameters and intermediate values. Exchangeability enables the method to partially evaluate
the network, and prune some computation on the fly.  The proposed method is architecture independent, thereby facilitating convenient adaptation and migration. The approach has been evaluated on image, graph and question answering tasks.

**Strengths:**

The main strength of the paper is that it is architecture independent. Thus, it is easily portable, likely to enable hassle free adaptation on new tasks. The exchangeability idea seems interesting (although  I am not fully convinced based on the given explanation). Finally, dynamic pruning has potential to reduce computation cost significantly for very large models, unlike static models.

**Weaknesses:**

In my opinion the paper has the following major issues:

1. Even though exchangeability sounds a good idea, I am bit skeptical about it relevance in the context of neural nets training. Once the training starts it is difficult to parameters are going to behave jointly and things get worse with the model size. Thus, I believe the assumptions underlying the  exchangeability for neural nets is hard to prove, even though, theoretically, in isolation "exchangeability" is reasonable.

2. Dynamic pruning has the problem of "resurrection". It is very difficult to predict, while the training is going on, which parameters are going to contribute when. Some parameters may look exchangeable with others or may look useless, but they may contribute later. The proposed approach is potentially detrimental for this.

3. I thin the experimental results are very weak in many aspects: i) choice of datasets, ii) choice of baselines, iii) experimental setup.  A work like this requires solid experimental evidence. Unfortunately the paper does not have it. It does not validate the claims made in the paper.

**Questions:**

In the weaknesses section, I have detailed the main issue. I do not any additional question that has significant importance.

---

> ### Author Response · Authors · 2025-11-21
> **Response**
>
> # Exchangeability in the context of training
>
> The parameters will surely behave jointly as training goes, and that is why we proved the exchangeability of them, which delineates the symmetric statistical relationship of them but does not imply statistical independence. Could you elaborate on which assumptions you referred to that are hard to prove? We are happy to answer any questions and address your concerns about our theory and proof.
>
> # Dynamic pruning has the problem of "resurrection"
>
> It seems like you misunderstood dynamic pruning. We would like to note that ExPrune makes pruning decisions on the fly for each inference forward pass, and ExPrune does not apply to, nor require changes to the training process. Maybe your message did not come across, as we find the terms you use odd and difficult to interpret. Could you elaborate more on this?
>
> # Evaluation
>
> We used standard baselines, datasets, and experimental setup in each problem domain. Could you elaborate on which part you found weak? We welcome your suggestions and are happy to incorporate them in the next revision of our paper.

---

> > ### Comment · Reviewer_KE6v · 2025-11-28
> > **Exchangeability**
> >
> > I am still not convinced about the "Exchangeability". It is bit too hypothetical and missing reasonably sound evidence in the context of neural net training. Although the authors claim it is "theory", I do not find any theory in it. Theory has no value unless it is a theory for something (in this case, neural net dynamics).

---

> > > ### Author Response · Authors · 2025-12-03
> > >
> > > > I am still not convinced about the "Exchangeability". It is bit too hypothetical and missing reasonably sound evidence in the context of neural net training. Although the authors claim it is "theory", I do not find any theory in it. Theory has no value unless it is a theory for something (in this case, neural net dynamics).
> > >
> > > Exchangeability is a statistical property that has been proposed by William Ernest Johnson in 1924, and has been extensively studied throughout the century, e.g., de Finetti’s theorem. In the context of neural nets and machine learning, it has been studied for quantifying the confidence range of neural network prediction [1], improving model performance for structured data [2], etc.
> > >
> > > On the topic of theory, ICLR has previously accepted (as oral papers) and spotlighted a lot of theory papers that do not “do something”, e.g., papers on learning theory. In our case, we use our theory to “do something” – partial computation – as described in our paper.

---

> > ### Comment · Reviewer_KE6v · 2025-11-28
> > **About resurrection**
> >
> > My point is that if the network training dynamics does not follow "Exchangeability", the resultant network is not going to be "Exchangeable". Thus, in my opinion, forcing "Exchangeability" condition on the final network and then use it to prune the network during inference does not seem convincing to me.

---

> > > ### Author Response · Authors · 2025-12-03
> > >
> > > > My point is that if the network training dynamics does not follow "Exchangeability", the resultant network is not going to be "Exchangeable". Thus, in my opinion, forcing "Exchangeability" condition on the final network and then use it to prune the network during inference does not seem convincing to me.
> > >
> > > We do not “force” any exchangeability condition on the neural network — we mathematically prove that it is naturally preserved by the training algorithms based on gradient descent, as shown in Theorem 2. This point should be abundantly clear in the paper, as acknowledged by other reviewers. This theoretical result is what allows for us to perform early termination without changing the neural network architecture or the training algorithm – i.e., we use our theoretical result “for something”.

---

> > ### Comment · Reviewer_KE6v · 2025-11-28
> > **Evaluation**
> >
> > This is the most critical part of my comment. Except CIFAR 100, other datasets are not very standard to test pruning efficacy.
> > The accuracy results are given as a passing remark.
> > Bigger benchmarks (something like ImageNet) would be the best choices for the effectiveness of pruning.

---

> > > ### Author Response · Authors · 2025-12-03
> > >
> > > > This is the most critical part of my comment. Except CIFAR 100, other datasets are not very standard to test pruning efficacy. The accuracy results are given as a passing remark. Bigger benchmarks (something like ImageNet) would be the best choices for the effectiveness of pruning.
> > >
> > > ogbg-molhiv is a standard benchmark in the OGB datasets used for graph neural networks [3, 4], and PIQA is a standard benchmark for LLMs [5, 6].
> > >
> > > Due to time and resource constraints, we have not run on even larger datasets. We want to note that we are the first to model the relationship among NN parameters with exchangeability and to perform partial computation during inference exploiting the property. We have demonstrated that the ExPrune algorithm generally works for 3 model families across 3 problem domains, with the largest model size containing 7.6B parameters, and the current paper is already packed with theory, algorithm, and a comprehensive evaluation. Scaling the algorithm to even larger models and larger datasets could be another interesting problem to be solved in another academic paper.
> > >
> > > [1] Fontana, Matteo, Gianluca Zeni, and Simone Vantini. "Conformal prediction: a unified review of theory and new challenges." Bernoulli 29, no. 1 (2023): 1-23.
> > >
> > > [2] Bloem-Reddy, Benjamin, and Yee Whye Teh. "Probabilistic symmetries and invariant neural networks." Journal of Machine Learning Research 21, no. 90 (2020): 1-61.
> > >
> > > [3] Xu, Keyulu, Weihua Hu, Jure Leskovec, and Stefanie Jegelka. "How powerful are graph neural networks?." International Conference on Learning Representations (2019).
> > >
> > > [4] Kipf, T. N. "Semi-supervised classification with graph convolutional networks." International Conference on Learning Representations (2017).
> > >
> > > [5] Gu, Albert, and Tri Dao. "Mamba: Linear-time sequence modeling with selective state spaces." In First conference on language modeling. 2024.
> > >
> > > [6] Jiang, Albert Q., Alexandre Sablayrolles, Antoine Roux, Arthur Mensch, Blanche Savary, Chris Bamford, Devendra Singh Chaplot et al. "Mixtral of experts." arXiv preprint arXiv:2401.04088 (2024).

---

### Official Review · Reviewer_SbWs · 2025-11-08

**Soundness:** 2
**Presentation:** 2
**Contribution:** 2
**Rating:** 2
**Confidence:** 3

**Summary:**

This paper develops a theoretical framework based on statistical exchangeability to reason about the distribution of neural network parameters and activations from initialization and throughout training. This framework is then used to motivate EXPRUNE, a dynamic weight pruning method that does not require model retraining and has general applicability.

In its current form the paper should be rejected because:
  1. The methodology is largely similar to previous methods with the main contributions being a new theoretical motivation using statistical exchangeability. However the argument for why exchangeability provides greater theoretical understanding than previous approaches is not made clearly or convincing.
  2. The overall experimental insight into performance of the method is lacking.
  3. Several important baselines are not considered.

**Strengths:**

The extension of statistical exchangeability to the distribution of neural network parameters is interesting and novel.

The top-1 prediction head method is novel and impactful. For transformer language models suitable for the edge deployment, the decoding head can be disproportionately FLOP intensive due to large vocabulary size relative to embedding dimension.

**Weaknesses:**

A few relevant prior work in dynamic pruning are not included in the literature review.
[1] uses traditional activation and weight magnitude pruning at test time to dynamically prune LLM weights.
[5] first performs the operation in low precision to predict negative outputs that will be dynamically pruned.
[6] performs the operation in decreasing order of significant bits.

There are also connections to the broader dynamic pruning/sparsity literature such as [11, 12, 13, 14] as well as other pruning methods that are not training free that are not made directly. This could be an opportunity to highlight that EXPRUNE is training and calibration free.

The experimental section does not have enough substance.
The paper only includes one major experiment conducted on 3 different model types (CNN, GCN, Transformer LLM) as well as on a statically pruned CNN.
Only an unoptimised baseline is used with the addition of SnaPEA for the first CNN experiment.
The paper would greatly benefit from a significantly expanded set of experiments, baselines and ablations.

In the main experiment, after a set of suitable hyperparameters is identified on the validation set. The subset containing the first 5 pareto slices is plotted. This is test set contamination.

**Questions:**

- Throughout the paper it is highlighted that the method does not require a specialized model or training algorithm, however it is not explicitly stated that retraining or calibration also is not required. A reader would benefit if this is emphasised early.
- Why was computation ordering not tested?
- Why was "ReLU Strikes Back" inference methodology (exploiting input sparsity) not used as a naive baseline with the normally trained ReLU models?
- Why is EXPRUNE not composed with exploiting input sparsity?
- With regards to baselines, direct comparison to other neuron level dynamic pruning methods such as [1, 4, 5, 6] would make the papers position in the literature more clear and highlight any unique benefits a statistical exchangeability based approach brings.
- The error free exact mode of SnaPEA was used in the experiments, however, a lossy predictive mode also exists. Why was it not used to show the FLOP vs accuracy trade off of the SnaPEA method?
- Does EXPRUNE also compose with activation aware static pruning methods? [2, 3, 7, 8, 9] What further insights are there to the importance of dynamic vs static activation information when pruning?
- Ideally there would be additional experiments demonstrating exchangeability in a practical setting would help bridge the exchangeability theory to the EXPRUNE method.
- Statistics of early ReLU classification for different computation order, check interval or other variables would all provide further insight in to the workings of the method and provide value to the reader.
- How does performance and early prediction ability vary across scales? [10]

I would raise my score if the experimental section was significantly expanded and more in depth and if I was convinced of a more fundamental connection between the statistical exchangeability framework and the EXPRUNE methodology.

Additional questions unrelated to the score:
- Is the computation order random or static?
- Can the top-1 prediction head be generalized to allow early sampling?

References:

[1] Koike-Akino et al. μ-MoE: Test-Time Pruning as Micro-Grained Mixture-of-Experts https://arxiv.org/abs/2505.18451

[2] Sun et al. A Simple and Effective Pruning Approach for Large Language Models https://arxiv.org/abs/2306.11695

[3] Frantar et al. SparseGPT: Massive Language Models Can be Accurately Pruned in One-Shot https://arxiv.org/abs/2310.04564

[4] Chen et al. CompRRAE: RRAM-based Convolutional Neural Network Accelerator with Reduced Computations through a Runtime Activation Estimation https://arxiv.org/abs/1906.03180

[5] Suresh et al. Early Prediction of DNN Activation Using Hierarchical Computations https://doi.org/10.3390/math9233130

[6] Ibrahim et al. DSLOT-NN: Digit-Serial Left-to-Right Neural Network Accelerator https://arxiv.org/abs/2309.06019

[7] Liu et al. AWP: Activation-Aware Weight Pruning and Quantization with Projected Gradient Descent https://arxiv.org/abs/2506.10205

[8] Hussien et al. Small Contributions, Small Networks: Efficient Neural Network Pruning Based on Relative Importance https://arxiv.org/abs/2410.16151

[9] Bhuiyan et al. Z-Pruner: Post-Training Pruning of Large Language Models for Efficiency without Retraining https://arxiv.org/abs/2508.15828

[10] https://huggingface.co/SparseLLM

[11] Yuan et al. Native Sparse Attention: Hardware-Aligned and Natively Trainable Sparse Attention https://arxiv.org/abs/2502.11089

[12] Shazeer et al. Outrageously Large Neural Networks: The Sparsely-Gated Mixture-of-Experts Layer https://arxiv.org/abs/1701.06538

[13] Raposo et al. Mixture-of-Depths: Dynamically allocating compute in transformer-based language models https://arxiv.org/abs/2404.02258

[14] Elhoushi et al. LayerSkip: Enabling Early Exit Inference and Self-Speculative Decoding https://arxiv.org/abs/2404.16710

---

> ### Author Response · Authors · 2025-11-21
> **Response part 1**
>
> # Highlighting ExPrune is Training and Calibration Free
>
> Thank you for the suggestions on writing. We will highlight that ExPrune is training and calibration free, and does not require architectural changes to the model in our next revision of the paper.
>
> # Relevant Prior Work
>
> Thanks for providing a list of works not included in our related work. We agree that the listed works have a connection to this work, and we will include them in our next revision of the paper. However, we think ExPrune and the listed works are not comparable due to the following:
>
> [1] is a LLM-specific technique and uses online calibration to prune the weights. In contrast, ExPrune is calibration free (as you pointed out) and general across a variety of model architectures. Although [1] is also online, it does not make pruning decisions in every forward pass as ExPrune does.
>
> [4, 5, 6] are similar to ComPreEND (cite [31] in our paper). These techniques require hardware/architectures that support bit slicing, and bit slicing itself also introduces overhead. They also only target convolutions. Some of them are also limited to a specific number representation. In contrast, ExPrune does not need bit slicing, can target a wide variety of model architectures besides convolutions, and works with any data types.
>
> As you pointed out, [11, 12, 13, 14] are not training free, and they also need architectural changes to the model (e.g., adding gating networks for MoEs, prediction heads for early exits). In contrast, ExPrune is training free and does not change the model architectures, enabling effortless application to trained models.
>
> Furthermore, ExPrune can potentially compose with the listed works. ExPrune could potentially be applied on top of [1] to further reduce computation in each model forward pass. ExPrune could reduce the number of MACs before applying [4, 5, 6] and reducing the number of bit computations per MAC. ExPrune could also be applied to the linear, convolution, transformer modules in the sparsified models in [11, 12, 13, 14]. We have presented a demonstration of applying ExPrune to statically pruned models. It is infeasible however, to include in the paper evaluation of the composition of ExPrune with every one of these techniques, as it requires a significant amount of resources and is beyond the scope of the paper. We would like to note, that the composition of techniques are not evaluated in the listed works.
>
> # Hyperparameter Tuning
> After we find a number of hyperparameters, we identify the 5 pareto slices evaluated on the validation set, not the test set, and plot their performance on the test set. We do not use the test set in the selection of the hyperparameters in any way, eliminating the possibility of test set contamination. We will make sure to revise the text to make this clearer in our next revision.
>
> # Why was computation ordering not tested?
>
> Could you elaborate on what you mean by testing computation ordering?
>
> # Composing with inference exploiting input sparsity
>
> Thank you for the suggestion. When counting the FLOPs/MACs, some prior works consider input sparsity and do not count when the input is zero, and some prior works do not consider input sparsity while counting. This depends on the assumption of the hardware/architecture—whether it supports skipping zero inputs and whether the mechanism brings performance benefits. In our experiment, we assumed that the hardware does not support skipping zero inputs but support skipping zero weights (in the experiment that composes ExPrune with statically pruned CNNs). In the future, we could also evaluate ExPrune assuming the hardware supports skipping zero inputs, like what the “ReLU strikes back” paper did.

---

> ### Author Response · Authors · 2025-11-21
> **Response part 2**
>
> # Comparing to SnaPEA
>
> Thank you for your suggestion. We would like to note that there is not enough information to reproduce the experiment. While a predictive mode of SnaPEA exists, its code and many details are not publicly available. We have shown that the lossless version of SnaPEA is much less performant than the lossless ExPrune. We would also like to note that SnaPEA requires a lot more branching per inference, which is bad for hardware acceleration, and it only natively supports CNNs without normalization layers and shortcut connections. We have extended SnaPEA to work with modern CNNs by fusing the batchnorm layers into the convolution layers and taking into account shortcut values. However, SnaPEA cannot be adapted to work with the GNN/LLM models.
>
> # Connection of the exchangeability theory to the ExPrune method
>
> By statistical exchangeability, we know that certain subnetworks’ partial outputs are from a shared latent probability distribution, with respect to random initializations of the network’s weights. This is why we can approximate the full network’s output with subnetworks’ partial outputs, and is the theoretical basis for the ExPrune algorithm. This is analogous to approximating the expected value of a distribution with the mean of samples drawn from the distribution. Without a shared distribution, such approximation would not be possible. Furthermore, confidence metrics for pruning, such as Wald’s statistics used in ExPrune explicitly requires exchangeability of the samples.
>
> Thank you for the suggestion on how we can highlight this more via experiments. If space allows, we will consider adding them in our next revision.
>
> # How does performance and early prediction ability vary across scales?
>
> Prior to OPT-7.6B, we have used the OPT-1.3B model for evaluation. We saw consistent results switching from the 1.3B model to the 7.6B model. In fact, the results are quite consistent even across model families in our experiment (~15% FLOPs reduction without accuracy loss in our CNNs, GNNs, LLMs). We will consider showing the benefits across different model sizes in our next revision.
>
> # Is the computation order random or static?
>
> The theory of exchangeability implies that any order would give the same accuracy and performance in expectation. In our experiment, we used a static order. We always compute the first 32 terms before determining the confidence via statistical tests or threshold check.
>
> # Can the top-1 prediction head be generalized to allow early sampling?
>
> Could you elaborate on what early sampling means in this context?

---

> ### Comment · Reviewer_SbWs · 2025-11-27
>
> Thanks for the response.
>
> # Computation order
>
> Line 281:
>
> "Computation order. $ξ_i$ can be computed in any order with the same expected accuracy due to their
> exchangeability. Optionally, they can be computed in certain heuristic orders"
>
> If the expected accuracy is the same regardless of the order, then why would one want to compute them
> in a certain heuristic order? Is this referring to ordering as mentioned in Sec 5.2 or the ordering used by SNAPEA?
>
> To summarize, my understanding of what you have presented:
>
> 1. If you take N models with different initializations and train them on the same data,
>    then across the N models the distributions of $ξ_i$ are exchangeable throughout training.
> 2. However, if you take one of those trained models and discard the others, and look at the distributions of $ξ_i$ within that model,
>    I am not convinced that those distributions are exchangeable. I.E. in the case of the CNN, GCN, LLM models.
>
> The second part is why I am struggling to see the connection between exchangeability and the proposed method.
> It seems to me that the method could have been derived as a generalisation of some of the existing methods I referenced,
> without appealing to exchangeability.
>
> # Broader experiments
>
> I agree that you are not expected to cover all possible experiments and compositions with previous methods.
> I am happy for additional compositional experiments to be passed over entirely.
>
> However, I am not convinced that the experiments you have presented are sufficient to demonstrate the generality of the method.
>
> The experiments can be summarised as follows:
>
> 1. Toy experiment with MNIST on 2 layer MLP showing exchangeability. (Fig. 2,3)
> 2. Performance of EXPRUNE against unoptimized baseline on CNN, GCN, LLM (Fig. 5)
>    - Fig 5a: Compared against lossless SNAPEA.
> 3. Performance of EXPRUNE against unoptimized baseline on CNN when composed with 1.93%, 1.74% and 1.83% dense statically pruned models (Fig. 6)
>
> Points:
>
> 1. The toy experiment is useful to illustrate exchangeability, but it is not clear how this translates to a real world scenario.
>    Surely similar plots could have been generated of the CNN, GCN and LLM models specifically showing the exchangeability of $ξ_i$ in practice?
>    I am happy for the MLP example to to show the distribution over many training runs, however want to see evidence that this holds in practice for the models of interest for fixed parameters with real world input data distribution.
> 2. I accept that the extent to which prior work can be compared is limited to multiple reasons (e.g. code availability, generality of method, etc...).
>    However, I still do not think that sufficient attempt has been made to compare against the prior work.
>    In cases where prior work was not general, EXPRUNE could have been applied to their existing baseline experiments to show the improvement.
>    In the cases where prior work was hardware specific, the methods could still have been compared in simulation to at least provide some reference point.
> 3. The presentation of these results is odd.
>    1. Fig6a 1.93%; 6b 1.74%; 6c 1.93%, 1.74% and 1.83% combined.
>    2. Why not show all three individually in Fig 6 or only all three combined as in Fig 6c?
>    3. Additionally, there is no baseline reported for EXPRUNE for 100% dense VGG CNN. Considering the threshold method seems to work better with static pruned VGG than dense ResNet, it would be useful to see how EXPRUNE compares on dense VGG as well.
>
>
> # Can the top-1 prediction head be generalized to allow early sampling?
>
> If top-k sampling can be performed instead of argmax (top-1) prediction, to allow for more diversity in the sampled sequences in the case of autoregressive LLM inference.

---

> > ### Author Response · Authors · 2025-12-03
> >
> > Thanks for the further questions.
> >
> > # Computation order
> >
> > > If the expected accuracy is the same regardless of the order, then why would one want to compute them in a certain heuristic order? Is this referring to ordering as mentioned in Sec 5.2 or the ordering used by SNAPEA?
> >
> > It is referring to the ordering mentioned in Sec 5.2. Though the expected accuracy does not depend on the ordering, the amount of computation does in the case of sparse (statically pruned) models. This is because certain $\xi_i$’s are “cheaper” to compute than others when skipping zero (pruned) weights. In Section 5.2, we compute the cheap $\xi_i$’s first.
> >
> > > Connection of the theory of exchangeability to ExPrune algorithm
> >
> > We first want to clarify that the random space in which we derive the exchangeable distributions of $\xi_i$’s is over random model initialization, and thus there is no randomness/distributions/exchangeability defined for just a single model. We explain why exchangeability still matters for the proposed ExPrune algorithm below.
> >
> > The theory of exchangeability informs us that certain subnetworks’ partial outputs are drawn from a shared latent probability distribution. This is why we can approximate the full network’s output with subnetworks’ partial outputs, and is the theoretical basis for the ExPrune algorithm. This is analogous to approximating the expected value of a distribution with the mean of samples drawn from the distribution. Without a shared distribution, such approximation would not be possible. Furthermore, confidence metrics for pruning, such as Wald’s statistics used in ExPrune explicitly requires exchangeability of the samples. Consider if each $\xi_i$ is drawn from a distinct distribution, then it is impossible to infer any information about some $\xi_i$’s given some others, and the Wald’s test is ill-formed.
> >
> > # Broader experiments
> >
> > > Surely similar plots could have been generated of the CNN, GCN and LLM models specifically showing the exchangeability of $\xi_i$’s in practice?
> >
> > We want to note that this figure was generated only for pedagogical purposes — we have mathematically proved the exchangeability of $\xi_i$’s in theorem 2 and it applies to all models. We could generate similar plots for other models but it would involve a gigantic amount of compute resources for retraining the models (including the LLMs) for 500 times, which is both unrealistic and unnecessary. A perhaps not accurate analogy is that the Pythagorean theorem, once mathematically proved, holds for all triangles, and people do not need to empirically test that it holds for triangles of different sizes, and it is unrealistic to do so since there are infinite number of triangles of different sizes.
> >
> > > In cases where prior work was not general, EXPRUNE could have been applied to their existing baseline experiments to show the improvement. In the cases where prior work was hardware specific, the methods could still have been compared in simulation to at least provide some reference point.
> >
> > For the first case you mentioned, we did compare against SnaPEA on the CNNs in our best effort (we had to adapt the CNNs model architecture and also the SnaPEA algorithm to make it work), and we saw a clear improvement in Figure 5. For the second case you mentioned, even in simulation, one needs to design a suitable compute architecture for the algorithm, which in itself could be a sufficient contribution for another academic paper.
> >
> > > Why not show all three individually in Fig 6 or only all three combined as in Fig 6c?
> >
> > We show the performance of ExPrune on specific statically pruned models to make a point that “ExPrune can provide additional FLOPs reduction even in heavily statically pruned models”. We show the performance of ExPrune in all statically pruned models combined to make a point that “ExPrune can compose with static pruning to achieve greater FLOPs reduction than static pruning alone”, which is demonstrated by the fact that the ExPrune points in blue and black are on the Pareto frontier and dominate the only-static-pruning points in red.
> >
> > # Can the top-1 prediction head be generalized to allow early sampling?
> >
> > That is an interesting point. We think ExPrune could be extended from top-1 to top-k sampling by adapting how the confidence test is performed, i.e., testing whether the margin of top-k to others is large enough to terminate early.

---

### Comment · Area_Chair_Ng3R · 2025-11-27

Dear Reviewers,

Could you please consider the author responses, and reply if you have not already.

Thank you.

AC

---

### Meta-Review · Area_Chair_CUSS · 2025-12-08

**Summary:**

The paper proposes EXPRUNE, a dynamic pruning method grounded in the statistical concept of "exchangeability." The authors suggest that certain neural network parameters and intermediate values satisfy exchangeability, which allows these parameters to be permuted, allowing for partial computation, followed by statistical tests to prune remaining parameters. The method is evaluated across CNNs (ResNet), GNNs (GCN), and Transformers (OPT). However, reviewers express concerns about both the scope of the experiments, with concerns about missing benchmarks (datasets) and baselines (prior work) and over-reliance on FLOPS as a metric, as well as the lack of theory. In particular, the paper does not formally prove that exchangeability is preserved after subsequent model optimization/reduction steps, such as finetuning, thus lacking an end-to-end guarantee.

**Reviewer Concerns:**

Although specific questions about intuition or design choices were partially addressed, I believe the major concerns about experiments and theory are all still outstanding.

**Reviewer Scores:**

Given that multiple reviewers responded with multiple concerns in the discussion phase and certain reviewers explicitly declared their score would not be changed at the time, it would be difficult for me to believe that reviewers would have increased their scores.

---

### Decision · Program_Chairs · 2026-01-26

Reject